# The Stiffness of the Ascending Aorta Has a Direct Impact on Left Ventricular Function: An In Silico Model

**DOI:** 10.3390/bioengineering11060603

**Published:** 2024-06-12

**Authors:** Wolfgang Anton Goetz, Jiang Yao, Michael Brener, Rishi Puri, Martin Swaans, Simon Schopka, Sigrid Wiesner, Marcus Creutzenberg, Horst Sievert, Ghassan S. Kassab

**Affiliations:** 1Cardiothoracic Surgery, University Hospital Regensburg, 93053 Regensburg, Germany; wolfgang1.goetz@klinik.uni-regensburg.de (W.A.G.); marcus.creutzenberg@ukr.de (M.C.); 2Dassault Systèmes, Johnston, RI 02919, USA; 3Division of Cardiology, Columbia University Irving Medical Center, New York, NY 10027, USA; mib2102@cumc.columbia.edu; 4Cleveland Clinic, Cleveland, OH 44195, USA; 5St. Antonius Ziekenhuis, 3435 Nieuwegein, The Netherlands; m.swaans@antoniusziekenhuis.nl; 6CardioVascular Center, 60389 Frankfurt, Germany; horst@sievert.md; 7California Medical Innovations Institute, San Diego, CA 92121, USA

**Keywords:** in silico, finite element method, computational simulation, aortic stiffness, atrioventricular plane displacement, ventricular strain, ventricular function, HFpEF

## Abstract

During systole, longitudinal shortening of the left ventricle (LV) displaces the aortic root toward the apex of the heart and stretches the ascending aorta (AA). An in silico study (Living Left Heart Human Model, Dassault Systèmes Simulia Corporation) demonstrated that stiffening of the AA affects myocardial stress and LV strain patterns. With AA stiffening, myofiber stress increased overall in the LV, with particularly high-stress areas at the septum. The most pronounced reduction in strain was noted along the septal longitudinal region. The pressure–volume loops showed that AA stiffening caused a deterioration in LV function, with increased end-systolic volume, reduced systolic LV pressure, decreased stroke volume and effective stroke work, but elevated end-diastolic pressure. An increase in myofiber contractility indicated that stroke volume and effective stroke work could be recovered, with an increase in LV end-systolic pressure and a decrease in end-diastolic pressure. Longitudinal and radial strains remained reduced, but circumferential strains increased over baseline, compensating for lost longitudinal LV function. Myofiber stress increased overall, with the most dramatic increase in the septal region and the LV apex. We demonstrate a direct mechanical pathophysiologic link between stiff AA and reduced longitudinal left ventricular strain which are common in patients with HFpEF.

## 1. Introduction

During systolic longitudinal shortening of the left ventricle (LV), the atrioventricular plane is displaced towards the LV apex [1,2]. This displacement of the aortic root by 16 mm (range 14 to 19 mm) [1,2,3] produces longitudinal stretching of the ascending aorta (AA) by 11.6 ± 2.9 mm (range: 3 to 19 m), which does not differ between males and females (11.8 ± 2.9 mm vs. 11.2 ± 2.9 mm) [4], while the aortic arch and the apex are only displaced by 2.9 ± 0.4 mm (range 0 to 6 mm) [4] and 1.9 ± 0.5 mm (range −0.1 to 5.1 mm), [3] respectively. The force required to produce a longitudinal stretch of the AA represents a direct mechanical load on the LV that may have important implications for the relationship between aortic stiffness and LV systolic longitudinal function [5].

Atrioventricular plane displacement and LV longitudinal shortening are the primary contributors to heart contraction, accounting for 60% of the LV stroke volume and 80% of the right ventricular (RV) stroke volume [6]. Since LV longitudinal shortening is the major contributor to LV stroke volume [3], alterations in AA elasticity and subsequently increased mechanical load on the LV may play an important role in heart failure, particularly in heart failure with preserved ejection fraction (HFpEF) where reduced longitudinal strain and the significant impairment of longitudinal LV systolic function is common [7,8,9,10,11,12,13].

The fact that HFpEF symptoms strongly correlate with increased arterial stiffness [5,14,15,16,17] suggests a possible pathophysiologic link between AA stiffness, reduced AA stretching, decreased atrioventricular plane displacement and alterations in LV systolic longitudinal function, contributing to the pathophysiology of HFpEF syndrome [5,12,15].

The present computational analysis was undertaken to mechanistically qualify and quantify the effects of reduced AA longitudinal stretchability upon LV mechanics, an interaction which was not yet studied. We expect that these insights can potentially lead to better understanding of the HFpEF complex and might lead to novel therapeutic approaches with the intent to restore normal LV geometry and function.

## 2. Methods

### Computational Model

We utilized the Living Left Heart Human Model by Dassault Systèmes Simulia Corporation (LLHH), capable of simulating LV performance, pressure–volume loops and stress and strain analyses, all correlating with clinical observations [18,19]. The finite element model included the AA, LV, left atrium, mitral valve, aortic root and pericardium. The dynamic response was governed by realistic structural and blood flow physics, and the heart contraction was driven by electrical excitation. Blood was represented using a combination of three-dimensional hydrostatic fluid cavities for the heart chambers and system-level chambers to represent arterial and pulmonary compliances. Blood flow occurred inside a closed-loop system between the chambers and the circulatory system through fluid link elements. Details of the model can be found under the “Simulation” and “Virtual Human” sections at Dassault Systèmes User Assistance, located at http://help.3ds.com (accessed on 11 June 2024).

The passive material response of the cardiac tissue used an anisotropic hyper elastic formulation proposed by Holzapfel and Ogden, as described in Equation (1) [20].

The passive material parameters were calibrated as follows: the biaxial and triaxial experimental data published by Sommer et al. [21] were used for initial calibration, and diastolic filling tests were used to augment the calibration of the eight material parameters, a, b, a_f_, b_f_, b_s_, a_s_, b_s_, a_n_ and b_n_, which describe the ventricular passive material properties based on the methods described in Klotz et al. [22,23] (Table 1).
(1)$$\Psi_{dev}=\frac{a}{2b}\exp[b\left(I_{1}-3)\right]+\sum_{i=f,s}\frac{a_{i}}{{2b}_{i}}\left\{exp\left[b_{i}\left({{(I}_{4i}-1)}^{2}\right)\right]-1\right\}+\frac{a_{fs}}{{2b}_{fs}}[exp(b_{fs}I_{8fs}^{2}-1)]$$

Equation (1): Passive material response of cardiac tissue. Ψ_dev_ is the deviatoric strain energy. a, b, a_f_, b_f_, b_s_, a_s_, b_s_, a_n_ and b_n_ are ventricular passive material parameters.

The active tissue response contains length-dependent considerations of regional sarcomere lengths, affecting the stress components in the fiber and sheet directions in the constitutive model. The active-tissue material model intended to capture the Frank–Starling effect (i.e., the strength of the heart’s systolic contraction is directly proportional to its diastolic expansion) [24]. The active contraction was simulated by adding stress in the direction of the muscle fiber defined by a time-varying model of elastance [25] and followed
(2)$$\sigma_{af}\left(t,E_{ff}\right)=\frac{T_{max}}{2}\frac{{Ca}_{0}^{2}}{{Ca}_{0}^{2}+E{Ca}_{50}^{2}\left(E_{ff}\right)}\left(1-cos\left(\omega \left(t,E_{ff}\right)\right)\right),$$
where
$$E{Ca}_{50}^{2}\left(E_{ff}\right)=\frac{{{Ca}_{0}}_{max}}{\sqrt{e^{B\left(l\left(E_{ff}\right)-l_{0}\right)-1}}}$$
$$\omega \left({t,E}_{ff}\right)=\left\{\begin{array}{c} \pi \frac{t}{t_{0}}\;when\;0\leq t\leq t_{0}\\ \pi \frac{t-t_{0}+t_{r}\left(l\left(E_{ff}\right)\right)}{t_{r}}\;when\;t_{0}\leq t\leq t_{0}+t_{r}\left(l\left(E_{ff}\right)\right)\\ 0\;when\;t\geq t_{0}+t_{r}\left(l\left(E_{ff}\right)\right) \end{array}\right.$$
$$t_{r}\left(l\right)=ml+b$$
$$l\left(E_{ff}\right)=l_{r}\sqrt{2E_{ff}+1}$$

T_max_ (N/mm^2^) is a scalar factor for myocardial contractility that represents the isometric tension achieved at the longest sarcomere length and maximum peak intracellular calcium concentration. Ca_0max_ is the peak intercellular calcium concentration. B governs the shape of peak isometric tension and sarcomere length relation. l_0_ is the sarcomere length below which no active force develops. l_r_ is the initial sarcomere length. t_0_ is the time needed to reach the peak tension. m and b are coefficients that govern the relationship between the linear relaxation duration and sarcomere length. E_ff_ is a Lagrangian strain tension component aligned with the local muscle fiber direction [25].

We simulated the mechanical constraints imposed by the pericardium by applying the physiological boundary condition on the ventricular epicardium to achieve realistic atrioventricular plane motion and radial inward motion of the epicardium as described in humans [3]. Forty-nine clusters of nodes, evenly distributed on the epicardium surface, were constrained via a spring with higher stiffness when closer to the apex and lower stiffness when closer to the base [26].

The heart was constrained via boundary conditions at the cut planes of the aortic root and pulmonary veins. Each cut plane was constrained relative to a central reference point, and the reference point of the pulmonary veins was fixed. The aortic root was constrained from rotation but allowed to stretch. Aortic elasticity was modeled via a spring representing the AA stiffness. The stiffness of the spring was initially set at 0.5 N/mm at baseline to achieve a realistic translation of the proximal aorta of 11.0 mm during systole [27]. The spring stiffness was increased to 10 N/mm to model a stiff AA until a stationary aorta (stationary plane of the sino-tubular junction) was achieved. We performed three simulations under the following conditions: (A) The effect of a mobile AA using normal AA stiffness as elastic spring stiffness to constrain the aortic root motion. (B) The effect of stiffening the AA by immobilizing the aortic root at the sino-tubular junction. (C) In response to the restricted AA motion and consequently reduced cardiac output, sympathetic nerves and humoral regulation are expected to trigger an increase in myocardial contractility to restore normal cardiac output. Accordingly, we increased myocardial contractility T_max_ from 0.2 N/mm^2^ to 0.4 N/mm^2^ to achieve the same stroke volume and effective stroke work (area under the pressure–volume loop) as at baseline. Myocardial strain was calculated as the relative length change between the diastolic state and the systolic states. The LV strains were measured along the radial, circumferential and longitudinal directions at twelve locations (three axial and four circumferential locations) at both the epicardium and endocardium. The averages of tensile strains were reported with positive values, and compressive strains were reported with negative values and depicted as bar graphs. Baseline values were in the reported range of normal human LV strains [28]. Volumetric-averaged myofiber stress was calculated at end-systole in MPa (N/mm^2^) and presented as a contour plot in LV parasternal long-axis cut planes. Left ventricular pressures and volumes were computed and depicted as pressure–volume loops. The area under the pressure–volume loop represents the total effective work (Joule) generated by ventricular contraction as follows:(3)SW=SV ∗ MAP

Stroke work (SW) is the area under the pressure–volume loop, determined by the product of stroke volume (SV) and Mean Arterial Pressure (MAP).

## 3. Results

### 3.1. Baseline Simulation

At baseline simulation with contractility T_max_ of 0.2, the aortic root moved 11.0 mm towards the apex during systole, while the apex remained stationary. The baseline simulation computed a stroke volume of 92.2 mL and a stroke work of 8748 Joules (Table 2). The calculation demonstrated a typical pressure–volume loop (Figure 1A). The strain profiles of the LV at end-systole are presented in Figure 2 and Figure 3 and Table 3. During systolic contraction, the wall thickened with a radial strain of 0.63 ± 0.11; the circumference reduced with a circumferential strain of−0.20 ± 0.05 (Table 3, Figure 2); the average apex base length shortened with a longitudinal strain of −0.16 ± 0.01 (Table 4, Figure 3). The calculated average myofiber stress was 0.056 ± 0.036 MPa (Table 5). The systolic regional stress distribution at end-systole is presented as a contour plot in Figure 4A. The long-axis profile, with the profile of the LV at end-diastole as a gray silhouette, is represented as a contour plot in Figure 5A. The highest myofiber stress areas appeared at the mitral annulus, the fibrous trigones, the aorto-mitral junction and the papillary muscle tip.

### 3.2. Effect of Baseline Contractility with a Stiff Ascending Aorta

For simulating a stiff AA, the aorta was immobilized at the sino-tubular junction in a simulation with baseline contractility (T_max_ 0.2) (Figure 4B and Figure 5B). The transverse end-systolic diameter at the center of the longitudinal LV axis at the tip of the papillary muscles was reduced from 59 mm to 57 mm or 3.4% below the baseline value (Figure 6A,B). The cross-sectional profile of the LV shape demonstrated that LV tended to be more ovalized at ES. Compared to baseline measures, end-diastolic pressure was increased by 8.5%, and pressure–volume loop analysis (Figure 1A) showed end-systolic LV pressure was reduced by 9.1%, stroke volume was reduced by 10.9% and effective stroke work was reduced by 19.0% (Table 2). The strain profiles of the LV at end-systole are presented in Figure 2, Figure 3 and Table 3. The average radial strain was reduced by 20.2 ± 2.4% compared to the baseline strains. The circumferential strain was reduced by 6.8 ± 10.9%, and the average longitudinal stain was reduced by 48.4 ± 36.9% (Table 3, Figure 2), while the septal longitudinal strain was reduced greatest by 94.1%. The anterior, lateral and posterior strain measures were reduced by 41.2%, 13.3% and 40.0% respectively, indicating that AA stiffening exerts the greatest effect upon septal longitudinal stain (Table 3, Figure 3). The average myofiber stress increased by 37% compared to the baseline from 0.056 ± 0.036 to 0.076 ± 0.042 MPa (Table 5). The systolic regional stress distribution at end-systole is presented as a contour plot in Figure 4B. The long-axis profile of regional stress distribution with the LV profile at end-diastole is presented as a gray silhouette in Figure 5B. Stress increased overall in the LV, with very high stress areas noted at the septum, the papillary muscles, the mitral annulus, the fibrous trigones and the aorto-mitral junction. The comparison of the long-axis contour at end-systole with the gray silhouette at end-diastole indicates a reduced displacement of the aorto-mitral junction and the lateral mitral annulus to the apex of the LV (Figure 5B).

### 3.3. Effect of Increasing Myocardial Contractility with a Stiff Ascending Aorta

In order to restore similar stroke volume (Table 2) in the model with a stiff AA, T_max_ was increased from T_max_ 0.2 N/mm^2^ to 0.4 N/mm^2^ (Figure 3C and Figure 4C), resulting in a nearly identical pressure–volume loop, stroke volume and effective stroke work compared with baseline (Figure 1B). End-diastolic left atrial pressure decreased from 8.5% to 0.7% and end-systolic pressures increased from −9.1% to −1.3% of baseline values (Table 2). The transverse end-systolic diameter at the center of the longitudinal LV axis at the tip of the papillary muscles was reduced from 59 mm to 55 mm or 6.8% below the baseline value (Figure 6C). The cross-sectional profile of the LV demonstrated increasing end-systolic ovalization of the LV shape.

Compared to baseline strains, the average radial and longitudinal strain was reduced by 3.7 ± 8.8% and 37.5 ± 35.0%, respectively, while circumferential strain increased by 13.6 ± 29.1% over baseline (Table 3, Figure 2). The increased radial strain compensated for radial and longitudinal strain loss to achieve a similar stroke volume as at baseline (Figure 1B). All reginal longitudinal strains remained reduced in comparison to baseline, with septal, anterior, lateral and posterior strains being reduced by 82.4%, 23.5%, 6.7% and 33.3%, respectively (Table 4, Figure 3). The comparison of the long-axis contour at end-systole with the gray silhouette at end-diastole indicates a persistent reduction in the displacement of the aorto-mitral junction and the lateral mitral annulus to the apex of the heart (Figure 5C). The calculated average myofiber stress increased from 0.076 ± 0.042 MPa to 0.090 ± 0.071 MPa and was, on average, 61.8 ± 88.3% higher in comparison to baseline (Table 5). The systolic regional stress distribution at end-systole is presented as a contour plot in Figure 4C and the long-axis profile of regional stress distribution with the profile of the LV at end-diastole as a gray silhouette in Figure 5C. Stress increased overall in the LV compared to the baseline and stiffened aorta with T_max_ 0.2 N/mm^2^, with a noticeable increase in stress along the septum in the middle at the longitudinal distance between the apex of the left ventricle to the aortic root, the septal left lateral wall and the papillary muscles. New high-stress regions also emerged at the right LV lateral wall and apex (Figure 4C and Figure 5C).

## 4. Discussion

During the cardiac cycle, the apex of the heart remains stationary since the heart cannot move out of the fluid-tight pericardial sac, which is tethered to the diaphragm. Additionally, the apex of the pericardium is connected to the caudal sternum via the sterno-pericardial ligament, interlinking the caudal sternum with the LV apex. This creates a relatively straight line of force between the stationary LV apex [6] at the caudal end and the aortic arch [4] at the cranial end with the aortic root in between that propels up and down by (16.4 ± 0.5 mm) during the cardiac cycle, when the LV shortens and the AA gets stretched [3,6]. Consequently, stiffening of the AA is expected to negatively affect this line of force reaching from the epicardial apex to the AA [29].

The present computational simulation demonstrates that stiffening of the AA increases myocardial stress and affects LV strain patterns. Myofiber stress increased overall in the LV with high-stress areas noted at the septum, the papillary muscles, the mitral annulus, the fibrous trigones and the aorto-mitral junction. The most pronounced reduction in strain was noted along the septal longitudinal region, the direct connection between the LV apex and the aortic root. This is of particular relevance since LV longitudinal shortening is a major contributor to heart contraction, accounting for 60% of the LV stroke volume and 80% of the right ventricular (RV) stroke volume [6]. The pressure–volume loops revealed that aortic root stiffening caused a deterioration in LV function with increased end-systolic volume, reduced systolic LV pressure, decreased stroke volume and effective stroke work with elevated end-diastolic pressure.

When increasing myofiber contractility, the pressure–volume loops indicated that stroke volume and effective stroke work could be recovered, as it allowed an increase in LV end-systolic pressure and a reduction in end-diastolic pressure. Longitudinal and radial strains remained reduced, but the circumferential strains increased over baseline, compensating for lost longitudinal LV function. But myofibers demonstrated increased overall stress at the same time. The most dramatic increase in myofiber stress was found in the septal region, the apex of the LV and in the papillary muscles. When the papillary muscle contracts, the mitral valve secondary chordae tendineae pull the center of the atrioventricular plane, including the aortic root at the aorto-mitral curtain, towards the LV apex. This action supports the aorto-mitral angle and the displacement of the aortic root in systole [30] which explains the increased stress observed at the fibrose trigones and the papillary muscles.

It is well recognized that arterial stiffening increases with age [5,14,31,32,33,34]. Multiple mechanisms have been proposed to explain age-dependent vascular stiffening, including alterations in endothelial function, structural protein composition, collagen crosslinking, geometric changes and neurohumoral signaling. Altered extracellular matrix architecture has been recognized as a key component of the pre-atherogenic state [14,33].

Aortic stiffening is known to cause reduced arterial compliance (elasticity), impaired aorto-ventricular interaction, pronounced reductions in longitudinal left ventricular function, reduced long-axis strain, impaired early diastolic filling of the ventricle, higher left ventricular afterload and higher end-diastolic left atrial pressure [5,7,35,36].

In animals, a stiff aorta significantly increases myocardial oxygen consumption by 30% and the energetic costs for the heart for delivering a given stroke volume by 20–40% [37]. As a stiff ascending aorta would be displaced and stretched less than a compliant aorta would, the heart would have to contract with greater long-axis force to produce the same amount of aortic displacement and stroke volume [29].

It was demonstrated in humans that increased aortic stiffness is associated with reduced global longitudinal strain [5,14,15,16,17], and authors concluded that LV long-axis shortening and global longitudinal strain may be reduced when pulling against a stiffer aorta because of potential mechanical ventricular–vascular interaction. Consequently, when the aorta stiffens, the heart must contract with greater long-axis force in order to produce the same amount of aortic displacement [5]. This is consistent with our results showing that aortic stiffening imposes a direct mechanical load on the long-axis LV function.

This in silico study provides valuable insights into the influence of ascending aorta stiffness on the left ventricle function, focusing on conditions such as heart failure with preserved ejection fraction (HFpEF). The simulation results underscore the intimate mechanical coupling between the ascending aorta and the left ventricle and highlight a potential pathophysiological mechanism underpinning HFpEF. The simulations suggest that a stiffened ascending aorta may significantly affect key markers of cardiac function such as an increase in end-diastolic pressure and decreases in end-systolic pressure, stroke volume and effective stroke work. Furthermore, the geometric impact of aortic stiffness on the left ventricle manifests as a tendency for ovalization at end-systole, pointing to a reconfiguration of the cardiac architecture due to increased aortic stiffness. Strikingly, these alterations in LV function are also accompanied by distinct alterations in myocardial strain and stress parameters. These alterations include reduced tensile radial strain, compressive circumferential strain and longitudinal strain, as well as an increase in myofiber stress, particularly along the septum of the left ventricle. These results underscore the hypothesis of mechanical coupling between the ascending aorta and the left ventricle, where aortic stiffness may directly contribute to altered LV function.

While HFpEF is regarded in part as a disease of the vasculature characterized by augmented aortic stiffness causing unfavorable late-systolic afterload on the ventricle, we hypothesize that increased longitudinal myocardial stress or load due to the reduced axial elasticity of the AA might be the main mechanical cause of myocardial fatigue that will lead to altered left ventricular function and HFpEF symptoms [38,39]. Myocardial fatigue as a consequence of an adverse ventricular load is biologically plausible from a structural viewpoint as slow-twitch skeletal muscle and cardiac myocytes both share similar sarcomeric building blocks and the same β-myosin heavy-chain isoforms of striated muscle [40]. Although both are considered fatigue-resistant and mitochondria-rich cardiomyocytes which render the heart relatively fatigue-resistant, no matter how fatigue-resistant myofibers may be, it is reasonable to assume that there is a threshold of high workload at which myocardial fatigue may occur once its physiological buffers and metabolic capacity become overwhelmed and exhausted [40,41,42,43,44]. Such muscle fatigue provokes a release of myokines, e.g., interleukin-6, which trigger a downstream cascade of endothelial inflammation, mitochondrial dysfunction and oxidative stress. The consequent impairment of contractility and relaxation of a stressed but otherwise intact muscle fits well with the theory of myocardial inflammation in HfpEF [45,46]. If such a fatigued myocardium is left without the opportunity for recovery, a vicious cycle of increased myocardial stress, reactive myocardial hypertrophy and subsequent myocardial fatigue, rising ventricular end-diastolic/left atrial pressure and increasing wall stress eventually leads to HFpEF symptoms [12,40,41,45] (Figure 7).

The increased end-diastolic pressure observed in our model indicates potential progression towards HFpEF. Under such chronic stress (Figure 7) and concomitant inflammation, activated fibroblasts and myofibroblasts serve as central effectors in cardiac fibrosis [47,48], eventually leading to impaired relaxation and filling of the left ventricle, and consequently diastolic left ventricular failure.

This in silico study, although exploratory and hypothesis-generating in nature, provides a compelling argument for further investigations into the mechanical interplay between the ascending aorta elasticity and left ventricle function. Understanding the extent to which stiffness of the ascending aorta can influence ventricular function may lead to novel diagnostic and therapeutic strategies in managing heart failure. Given the high prevalence of aortic stiffness and impaired left ventricular longitudinal strain in patients with HFpEF, these findings provide a rationale for further research into therapies targeting ascending aorta stiffness and its downstream effects on ventricular function.

## 5. Limitation of Study

Although the Living Left Heart Human Model has seen considerable use in cardiac simulations, several limitations apply [18,24]. The LLHH model includes the aortic arch, left ventricle, left atrium, mitral valve, aortic root and pericardium, while the right heart is not captured in the model. As a result, potential effects or influences of these structures on the left heart were not accounted for. Furthermore, the material properties of the ascending aorta remained unaltered, and the stiff aorta was simulated by immobilizing the arch at the level of the sino-tubular junction, which may not have fully simulated the mechanics of a stiffened aortic wall.

In case of a stiff aorta, sympathetic nerves and humoral regulation are expected to increase myocardial contractility to restore normal cardiac output. However, hemodynamic feedback control was not modeled to automatically regulate myocardial contractility to maintain cardiac output. Rather, contractility was uniformly increased in all myocytes, neglecting possible anisotropy and remodeling of the left ventricle.

The chronic effect of a stiff ascending aorta with increased chronic workload on the myocardium and its consequences for the diastolic and systolic function of the left ventricle were not simulated in this model.

Noticeable surface irregularities (Figure 6) were a result of the simplified representation of the pericardium, achieved through springs connected to forty-nine clusters of nodes evenly distributed on the epicardium surface. Employing more nodes with spring stiffness inversely proportional to the displacement could potentially lead to a smoother surface but would not affect the overall results [26].

In summary, the LLHH model is a valuable tool for understanding the impact of stiffening of the ascending aorta on left ventricular function. However, it is essential to acknowledge its limitations, including the omission of certain heart structures, assumptions about material properties and the absence of hemodynamic feedback control and a simplified pericardium when interpreting the simulation results.

## 6. Conclusions

Our computational study demonstrated that the reduced axial elasticity of the ascending aorta imposes an additional load and stress upon the myofibers, which negatively affects longitudinal myocardial strain and subsequently deteriorates longitudinal LV function with increased end-diastolic pressure. Our simulations strongly suggest that ascending aorta stiffness is directly coupled to LV function and may potentially predispose patients to heart failure syndrome.

## Figures and Tables

**Figure 1 bioengineering-11-00603-f001:**
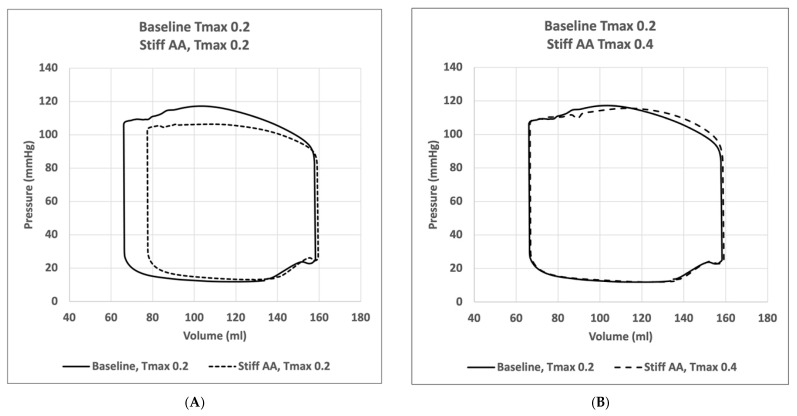
Pressure–volume loops. (**A**) Comparison of pressure–volume loop of left ventricle for simulation with mobile aorta (baseline) T_max_ 0.2 and with simulation with stiff AA T_max_ 0.2. (**B**) Comparison of pressure–volume loop of the left ventricle for simulation with mobile aorta T_max_ 0.2 and stiff AA T_max_ 0.4.

**Figure 2 bioengineering-11-00603-f002:**
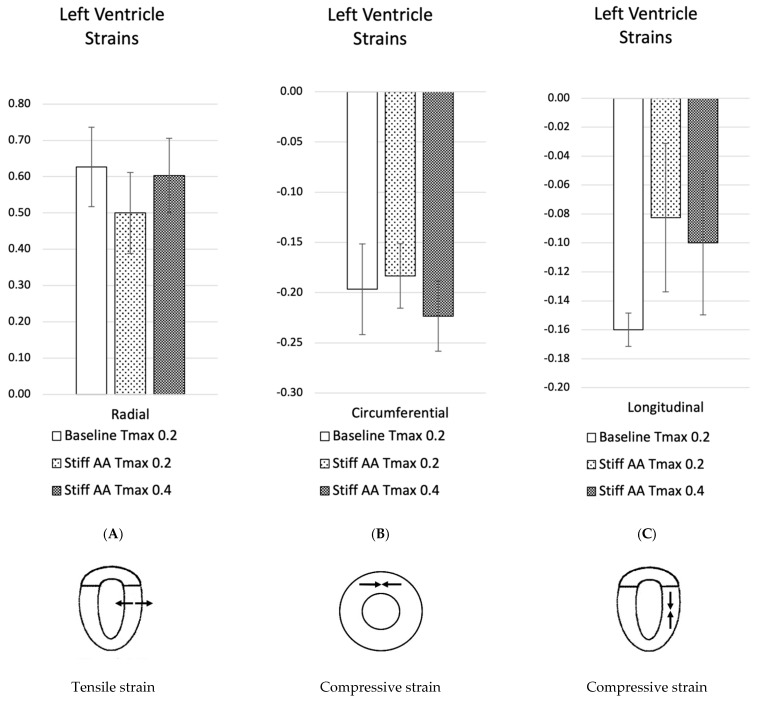
Left ventricular strain for baseline simulation T_max_ 0.2, simulation with stiff AA T_max_ 0.2 and simulation with stiff AA and increased contractility T_max_ 0.4. (**A**) Radial strain (three radial locations) is depicted as positive when there is wall thickening from diastole to systole. (**B**) Circumferential strain (three circumferential locations) is depicted as negative when circumference is reduced from diastole to systole. (**C**) Longitudinal strain (four longitudinal locations) is depicted as negative when apex base length is reduced from diastole to systole.

**Figure 3 bioengineering-11-00603-f003:**
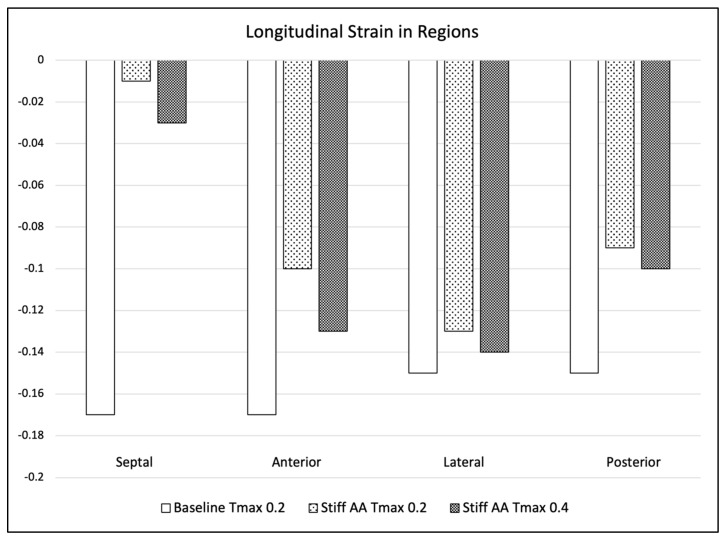
Longitudinal strains. Left ventricular longitudinal strain in four longitudinal regions (septal, anterior, lateral and posterior strain) at baseline T_max_ 0.2, stiff AAT_max_ 0.2 and stiff AA and increased myocardial contractility T_max_ 0.4.

**Figure 4 bioengineering-11-00603-f004:**
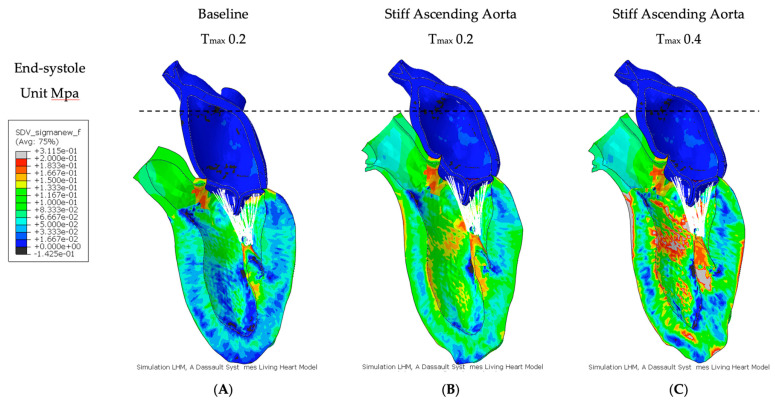
Myofiber stress. Long-axis profile of the LV at end of systole showing contours of myofiber stress. (**A**) Baseline Tmax 0.2, (**B**) stiff AA Tmax 0.2 and (**C**) stiff AA Tmax 0.4. Dotted line indicates baseline level of ascending aorta at end-diastole.

**Figure 5 bioengineering-11-00603-f005:**
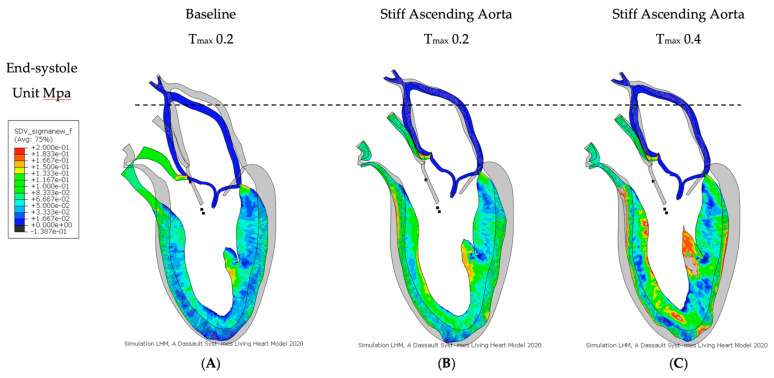
Myofiber stress. Long-axis profile of the LV at end of systole showing contours of myofiber stress. (**A**) Baseline Tmax 0.2, (**B**) stiff AA Tmax 0.2 and (C) stiff AA Tmax 0.4. The profile of the LV in diastole is placed behind the profiles of the cases (**A**–**C**) as a gray silhouette. Dotted line indicates baseline level of ascending aorta at end-diastole.

**Figure 6 bioengineering-11-00603-f006:**
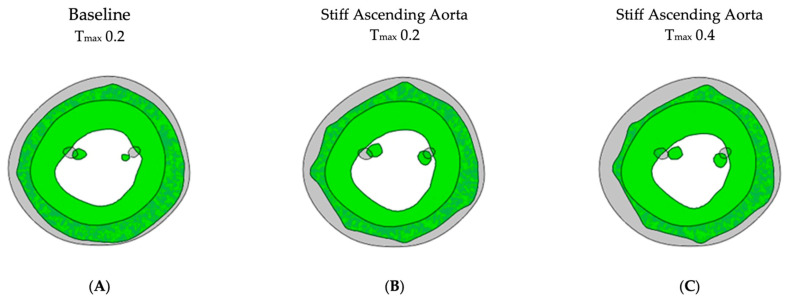
Left ventricular cross-sectional profile. Cross-sectional profile of the LV at center of the longitudinal LV axis and base of the papillary muscles. Gray color shows the end-diastolic shape at baseline; green color shows the end-systolic shape. (**A**) Baseline Tmax 0.2, diameter 59 mm, (**B**) stiff AA Tmax 0.2, diameter 57 mm and (**C**) stiff AA Tmax 0.4, diameter 55 mm.

**Figure 7 bioengineering-11-00603-f007:**
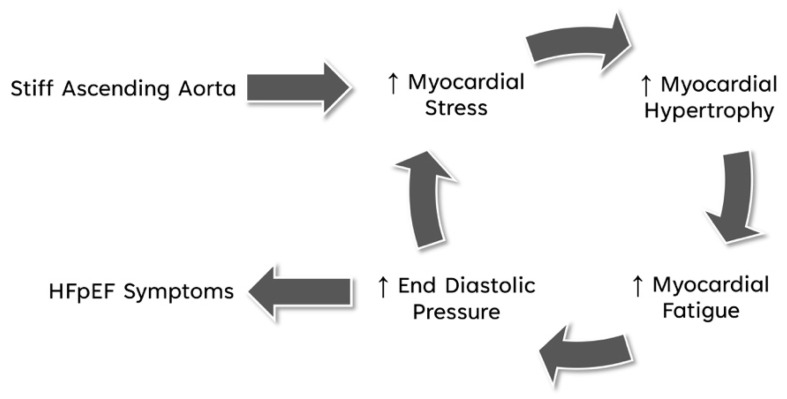
Vicious cycle of stiffened AA leading to HFpEF symptoms. Increased myocardial stress due to a stiff AA, reactive myocardial hypertrophy and subsequent myocardial fatigue, rising ventricular end-diastolic/left atrial pressure and increasing wall stress, which eventually leads to HFpEF symptoms.

**Table 1 bioengineering-11-00603-t001:** Constructive parameters for the passive and active material response.

Passive Parameters										
	a (MPa)	b	a_f_ (MPa)	b_f_	a_s_ (MPa)	b_s_	a_fs_ (MPa)	b_fs_		Calibration Data
Atrium	1.0 × 10^−3^	3.1	4.7 × 10^−3^	1.2 × 10^+1^	2.7 × 10^−3^	9.1	9.0 × 10^−7^	6.7 × 10^−4^		Sommer [21], Klotz [22]
Ventricle	3.9 × 10^−4^	3.7	1.9 × 10^−3^	1.4 × 10^+1^	1.1 × 10^−3^	1.1 × 10^+1^	3.6 × 10^−7^	7.8 × 10^−4^		Sommer [21], Klotz [22]
**Active Parameters**										
	**t0** **(s)**	**m** **(s/mm)**	**b** **(s)**	**l0** **(mm)**	**B** **(1/mm)**	**Ca0_max** **(mM)**	**Ca0** **(mM)**	**Tmax** **(Mpa)**	**Lr** **(mm)**	**Reference**
Atrium	0.05	1048.9	−1.5	0.00158	4750	4.35	4.35	0.1	0.00185	Sack [24], Guccione [23]
Ventricle	0.35	950	−1.5	0.00158	4750	4.35	4.35	0.2	0.00185	Sack [24], Guccione [23]

**Table 2 bioengineering-11-00603-t002:** Left ventricular pressures and volumes. Computational simulation of LV pressure and volume, stroke volume and effective stroke work at baseline T_max_ 0.2, stiff AA T_max_ 0.2 and stiff AA with increased contractility T_max_ 0.4 to recover stroke volume. (EDP: end-diastolic pressure; EDV: end-diastolic volume; ESP: end-systolic pressure; ESV: end-systolic volume; SVed-es: stroke volume; SW: stroke work).

	EDP	EDV	ESP	ESV	SVed-es	SW
	(mmHg)	(mL)	(mmHg)	(mL)	(mL)	(Joule)
Baseline T_max_ 0.2	11.85	158.30	117.10	66.10	92.20	8747.50
Stiff AA T_max_ 0.2	12.86	159.60	106.40	77.40	82.20	7084.50
Stiff AA increased contractility T_max_ 0.4	11.94	159.09	115.62	66.63	92.46	8794.81
Baseline vs. stiff AA T_max_ 0.2 vs. T_max_ 0.2	1.01	1.30	−10.70	11.30	−10.00	−1663.00
Baseline vs. stiff AA T_max_ 0.2 vs. T_max_ 0.2 (%)	8.52%	0.82%	−9.14%	17.10%	−10.85%	−19.01%
Baseline vs. stiff AA T_max_ 0.2 vs. T_max_ 0.4	0.09	0.79	−1.48	0.53	0.26	47.31
Baseline vs. stiff AA T_max_ 0.2 vs. T_max_ 0.4 (%)	0.74%	0.50%	−1.26%	0.80%	0.28%	0.54%

**Table 3 bioengineering-11-00603-t003:** Average radial, circumferential and longitudinal strain at baseline T_max_ 0.2, stiff AA T_max_ 0.2 and stiff AA and increased myocardial contractility T_max_ 0.4.

Average Strain	Radial	Circumferential	Longitudinal
Baseline T_max_ 0.2	0.63 ± 0.11	−0.20 ± 0.05	−0.16 ± 0.01
Stiff AA T_max_ 0.2	0.50 ± 0.11	−0.18 ± 0.03	−0.08 ± 0.05
Stiff AA T_max_ 0.4	0.60 ± 0.10	−0.22 ± 0.04	−0.10 ± 0.05
Baseline T_max_ 0.2 vs. stiff AA T_max_ 0.2	−0.13 ± 0.02	0.01 ± 0.02	0.08 ± 0.06
Baseline T_max_ 0.2 vs. stiff AA T_max_ 0.2 (%)	−20.21 ± 2.39%	−6.78 ± 10.86%	−48.44 ± 36.88%
Baseline T_max_ 0.2 vs. stiff AA T_max_ 0.4	−0.02 ± 0.06	0.03 ± 0.01	0.06 ± 0.06
Baseline T_max_ 0.2 vs. stiff AA T_max_ 0.4 (%)	−3.72 ± 8.78%	+13.56 ± 6.10%	−37.50 ± 35.00%

**Table 4 bioengineering-11-00603-t004:** Longitudinal strain in for regions, septal, anterior, lateral, posterior at baseline T_max_ 0.2, stiff AA T_max_ 0.2 and stiff AA and increased myocardial contractility T_max_ 0.4.

Longitudinal Strain	Septal	Anterior	Lateral	Posterior
Baseline T_max_ 0.2	−0.17	−0.17	−0.15	−0.15
Stiff AA T_max_ 0.2	−0.01	−0.10	−0.13	−0.09
Stiff AA T_max_ 0.4	−0.03	−0.13	−0.14	−0.10
Baseline T_max_ 0.2 vs. Stiff AA T_max_ 0.2	0.16	0.07	0.02	0.06
Baseline T_max_ 0.2 vs. Stiff AA T_max_ 0.2 (%)	−94.12%	−41.18%	−13.33%	−40.00%
Baseline T_max_ 0.2 vs. stiff AA T_max_ 0.4	0.14	0.04	0.01	0.05
Baseline T_max_ 0.2 vs. stiff AA T_max_ 0.4 (%)	−82.35%	−23.53%	−6.67%	−33.33%

**Table 5 bioengineering-11-00603-t005:** Average myofiber stress at baseline T_max_ 0.2, stiff AA T_max_ 0.2 and stiff AA with increased contractility T_max_ 0.4. Comparison with baseline stress.

Stress	Baseline	Stiff AA	Stiff AA
	T_max_ 0.2	T_max_ 0.2	T_max_ 0.4
	Baseline—Stroke Volume	Reduced—Stroke Volume	Recovered—Stroke Volume
(MPa)	0.056 ± 0.036	0.076 ± 0.042	0.090 ± 0.071
vs. baseline		36.98 ± 42.91%	61.76 ± 88.33%

## Data Availability

Data are available from the corresponding author upon reasonable request. Restrictions apply to the availability of some data, which were used under license for this study.

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
