# Peer review of "The Stiffness of the Ascending Aorta Has a Direct Impact on Left Ventricular Function: An In Silico Model"

_bioengineering, 2024, doi:10.3390/bioengineering11060603_

Round 1

Reviewer 1 Report

Comments and Suggestions for Authors

1. Introduction:

The introduction section could be improved by providing a more detailed discussion on the clinical impacts of aortic stiffness, not limited to HFpEF. Additionally, it is suggested to briefly review how these findings might influence current clinical practices or potential therapeutic interventions in the introduction. This could potentially provide a stronger motivation for the study.

2. Methods:

The description and validation of the model are relatively brief. It is recommended to include more detailed validation data against experimental or clinical data in the manuscript to enhance the reliability of the simulation results.

3. Results:

The description in the Results section is not detailed enough. It is recommended to provide clearer and more detailed descriptions for all the presented result figures. Additionally, the extensive use of technical terminology throughout the manuscript may inconvenience readers who are not familiar with computational biomechanics. It is advisable to include some explanations for parts of the content to make it more accessible.

4. Discussion:

The discussion on the potential mechanisms underlying the observed changes is insufficient in the manuscript. It is recommended to expand the discussion on the molecular or cellular responses that might be triggered, which could provide a more comprehensive perspective on the pathophysiology involved.

5. Limitations:

None.

6. Conclusion:

It is recommended to further clarify the thought process, succinctly review the main findings of the study, and clearly emphasize how these results meet the initial objectives of the research. Additionally, expand the discussion on the achievements made and consider their broader impact on future research, clinical practice, or potential treatment approaches.

Comments on the Quality of English Language

Author Response

Thank you for your time and efforts that will help us to improve our manuscript.

I appreciate your comments and I have revised our manuscript accordingly.

You will find attached the revised manuscript with highlights.

In Introduction the requested discussion was added.

In Methods Section 2.1 including the references describe in detail the computational model used.

In Discussion a discussion of potential mechanism underling the observed changes was added.

In Conclusion the requested discussion about impact and future research was added. Discussion about potential treatment approaches will follow in the subsequent manuscript.

Reviewer 2 Report

Comments and Suggestions for Authors

The paper presents a computational model to explore the impact of Stiffness of ascending aorta on left ventricular function. The presentation of the paper makes it difficult to identify the problem and understand the proposed model and the contribution of the study.

The abstract is not clear. The authors need to state briefly what has been done so far, what is the problem, what is the proposed solutions, and what are their findings.

The introduction section is short and does not convey the required background that the authors have solved. a brief description of the proposed method needs to be included.

The organization of the paper needs to be included at the end of the introduction section.

The related work section is missing. The authors need to include related work section that covers state-of-the-art methods along with the limitations of each. This will lead to problem identification and potential research gap.

Section 2.1 presents computational that consists of the discussion and equations taken from other sources. The contribution of this work is not clear.

This section is short and does not show the details of the methods (which is the title of the section).

The presentation of this section is not clear. From line 105 till the end of the section, the simulation setup and different parameters are presented in Tables. Why this content is part of this section?

The details of the experimental setup are missing in section 3.1. The details need to be included.

The conclusion section needs to cover discussion related to the findings of the study that justifies the title of the study.

Author Response

Thank you for your time and efforts that will help us to improve our manuscript.

I appreciate your comments and I have revised our manuscript accordingly.

You will find attached the revised manuscript with highlights.

As stated in the Abstract and Introduction the problem is:

LV longitudinal shortening stretches the Ascending Aorta. The effect of stiffening and consequent reduced stretchability of the aorta is expected to affect left ventricular stress and strain patterns. This was studied in silico.

This mechanical relation was not yet described.

We do not propose a model, it is a description of an in-silico study with defined conditions.

The contribution of the study is that we are the first demonstrating the direct mechanical pathophysiologic link between longitudinal stretch of the ascending aorta and longitudinal left ventricular strain.

Abstract

To our knowledge we are the first studying the effect of reduced longitudinal stretching on LV function in an in-silico model. We demonstrate a direct mechanical pathophysiologic link between reduced longitudinal Ascending Aorta elongation and reduced longitudinal left ventricular strain that is common in patients with HFpEF.

Introduction

There is no background that we intended to solve. We describe a phenomenon that was not studied before. Solving the problem i.e. restoring normal LV function in a situation with stiff ascending aorta is not the intention of this publication. A following manuscript will propose a solution. There is no related work and there is no state-of-the-art to solve the problem in this manuscript does not intend to propose any method or solution. The problem is a stiff ascending aorta with reduced stretchability (strain). The cause of the problem (stiff ascending aorta) and solution for the problem is not the intended task of this paper.

Methods

Section 2.1 describes in detail the computational model used, including references that validated the used computational model. Further elaboration and description of the Living Left Heart Human Model by Dassault Systèmes Simulia Corporation would be beyond the purpose of this publication. References allow for reviewing the computational model used.

The journal requests positioning of tables behind the first paragraph where the tables are referenced.

Results

The details of the experimental setup are described in detail in 2. Methods.

Conclusion

The title: “Stiffness of ascending aorta has a direct impact on left ventricular function: In silico model” is related to the conclusion: Our computational study demonstrated that the reduced axial elasticity of the ascending aorta (= stiffness of ascending aorta) imposes an additional load and stress upon the myofibers and negatively affecting longitudinal myocardial strain and subsequently deteriorating longitudinal LV function (=direct impact on left ventricular function) with increased end-diastolic pressure. Our simulations (=in-silico model) strongly suggest that ascending aorta stiffness is directly coupled to LV function and may potentially predispose patients to heart failure syndrome.

Reviewer 3 Report

Comments and Suggestions for Authors

The idea of bringing new evidence in the field of HFpEF is of great clinical interest, and the authors provide a mechanistic frame for a potential factor, namely aortic stiffness. Both HFpEF and aortic stiffness are largely debated in the literature, still the authors don't provide sufficient and relatively new background for the purpose of the experimental work, nor enough pathophysiological links between the two entities. The references should be improved with new evidences underlining the importance of the assessment of left ventricular workload in the presence of a rigid aorta.

The results are accurately presented, but the visual graphics could improve the understanding of the mechanisms. The limitations of the study are realistic and could open further field for investigation.

Author Response

Thank you for your time and efforts that will help us to improve our manuscript.

I appreciate your comments and I have revised our manuscript accordingly.

You will find attached the revised manuscript with highlights.

Introduction

We describe the isolated reduced longitudinal stretchability or strain of the ascending aorta, that imposes a higher load on the left ventricle. There is very little background and literature about this phenomenon. A few clinical studies were demonstrating in HFpEF reduced aortic descending or longitudinal strain and reduced left ventricular longitudinal strain. This studies we mention in 4 Discussion.

Beside the clinical observations we are not aware of any experimental or clinical study that elucidated this mechanical phenomenon.

To our knowledge there is no scientific publication about association between reduced longitudinal elasticity of the Ascending Aorta or reduced longitudinal strain of the ascending aorta and left ventricular function. To our knowledge all scientific work is related to general arterial or aortic stiffening and left ventricular function.

In Methods

In Discussion new research and insights regarding increased cardiac load and myocardial fatigue was added.

In Results Figrure 7 was added to improve understanding

Reviewer 4 Report

Comments and Suggestions for Authors

1. High Similarity Index: The manuscript has a reported similarity index of 71%, which is significantly high and raises serious concerns about the originality of the content. A detailed analysis shows that 38% and 26% of the content are copied from two primary sources, respectively. This level of duplication is unacceptable in scholarly publishing, as it indicates potential plagiarism and a lack of original contribution to the field.

2. Ethical Concerns: The high percentage of directly copied material without proper attribution or original analysis undermines the ethical standards required for publication. This not only questions the integrity of the manuscript but also the validity of the results and discussions presented.

3. Impact on Scientific Community: Publishing a paper with such a high degree of similarity to existing works could mislead readers and diminish the trust in the journal’s commitment to fostering original research. It also does not add any substantive new knowledge or insight to the existing body of literature.

Detailed Comments for the Authors:

1. The manuscript should significantly reduce its reliance on previously published materials. Any use of existing work must be appropriately cited and clearly distinguished from new contributions.

2. I recommend a thorough review of all sources to ensure all borrowed ideas, data, or text are properly attributed to their original authors.

3. Consider revising the manuscript to focus on providing unique insights or novel computational analyses that differentiate your work from the referenced sources.

4. Addressing these concerns is crucial before this work can be considered for publication. The current form of the manuscript does not meet the journal's standards for originality and ethical scholarly communication.

Given the significant issues with the similarity index and the ethical concerns it raises, I must recommend that this manuscript be declined. Revision and resubmission could be considered if the authors can significantly rewrite the content to ensure originality and proper attribution of sources. However, as it stands, the manuscript does not meet the necessary criteria for further consideration in the review process.

Author Response

Thank you for your time and efforts that will help us to improve our manuscript.

The very high similarities are with the authors previous pre-print of the same manuscript at Research Square on 28 August 2023

https://www.researchsquare.com/article/rs-3289570/v1

“Research Square lets you share your work early, gain feedback from the community, and start making changes to your manuscript prior to peer review in a journal.”

Round 2

Reviewer 2 Report

Comments and Suggestions for Authors

In response the previous comments, the authors have done quiet few changes in the article to address some of the comments.  The rest of the comments have been responded to in the response letter. The presentation can still be improved so that the other researchers can reproduce the same for further experimentation and research. 

Author Response

Dear Reviewer,

Thank you for acknowledging that we have addressed your comments and made significant improvements to the manuscript. I agree that there is always room for improvement in any manuscript. However, without specific feedback, it is challenging for us to identify areas where we could make further enhancements.

Best regards,

Wolfgang